# Pre-training Segmentation Models for Histopathology

**Payden McBee**[1]                                                    PM2KB@VIRGINIA.EDU

[1] *Department of Systems and Information Engineering, University of Virginia, Charlottesville, VA*

**Nazanin Moradinasab**[1]                                            NM4WU@VIRGINIA.EDU

**Sana Syed**[2]                                                       SS8XJ@VIRGINIA.EDU

[2] *Department of Pediatrics, University of Virginia, Charlottesville, VA*

**Donald E. Brown**[1]                                                 DEB@VIRGINIA.EDU

**Editors:** Under Review for MIDL 2023

## Abstract

In limited data settings, transfer learning has proven useful in initializing model parameters. In this work, we compare random initialization, pre-training on ImageNet, and pre-training on histopathology datasets for 2 model architectures across 4 segmentation histopathology datasets. We show that pre-training on histopathology datasets does not always significantly improve performance relative to ImageNet pre-trained weights for both model architectures. We conclude that unless larger labeled datasets or semi-supervised techniques are leveraged, ImageNet pre-trained weights should be used in initializing segmentation models for histopathology.

**Keywords:** Transfer learning, histopathology, segmentation.

## 1. Introduction and Related Works

Transfer learning is a technique where a model developed for a specific task can be reused as the initial model for a second task with limited labeled data. A common transfer learning approach for medical images is to start with the standard network architectures, e.g., VGG (Simonyan and Zisserman, 2014) and ResNet (He et al., 2016) pre-trained on the large-scale natural images such as ImageNet (Deng et al., 2009) and PASCAL VOC (Everingham et al., 2010), and then fine-tune them on medical images. The effectiveness of pre-trained deep convolutional neural networks (CNNs) with sufficient fine-tuning was investigated on four different medical imaging applications in Tajbakhsh et al. (2016). This study demonstrated that, in most cases, fine-tuning a pre-trained model achieved better performance and robustness in comparison to those trained from scratch. Similarly, Devan et al. (2019) demonstrated that transfer learning with ImageNet can significantly enhance model performance in detecting herpesvirus capsids in microscopy images, particularly when labeled data is limited. Conze et al. (2020) utilized a VGG-11 encoder pre-trained on ImageNet for the shoulder muscle MRI segmentation task. These results indicate that a CNN pre-trained on ImageNet learns features that are applicable to both natural and medical images. However, the gap in features between medical and natural images has motivated pre-training on medical datasets. Ray et al. (Ray et al., 2022) demonstrated an increase in performance and faster convergence for CNNs pre-trained on histopathology datasets relative to a model pre-trained on ImageNet. To the best of the authors' knowledge, the efficacy of utilizing a model pre-trained on natural images compared to a medical image pre-trained models for

nuclei segmentation tasks has not been investigated. Our contribution is to systematically compare segmentation performance using different pre-trained weights across 4 datasets derived from whole-slide-images: eosinophilic esophagitis (EoE) with 2,056 images, Crohn's disease (Crohns) with 800 images, Colorectal Nuclear Segmentation and the Pheno-types (CoNSeP) (Graham et al., 2019) with 660 images, and PanCancer Histology Dataset for Nuclei Instance Segmentation and Classification (PanNuke) (Gamper et al., 2019) with 7,901 images. We seek to answer the following: Does pre-training on histopathology datasets improve segmentation performance relative to encoders pre-trained on ImageNet?

## 2. Methods

For our analysis, we use HoVer-Net (Graham et al., 2019) and U-Net++ (Zhou et al., 2018) models. HoVer-Net has three separate task-specific decoders, which are used for nuclei detection, separation, and classification, respectively. U-Net++ has a single decoder to provide pixel classification. The Preact-ResNet50 is utilized as an encoder for both models. For HoVer-Net, we follow the exact hyper-parameters and training strategies presented in (Graham et al., 2019). For U-Net++, we train each model for 200 epochs and select the model that minimizes the validation binary cross-entropy (EoE and Crohns) or cross-entropy (CoNSeP and PanNuke) loss. We train and test each of the models for each of the 4 datasets given encoders with various pre-trained weights. We use the MoNuSAC ResNet50 encoder weights from (Graham et al., 2019), and the other weights are obtained by initializing a model with ImageNet and training it on a given histopathology dataset.

## 3. Experiments and Results

Table 1 shows the average performance of the U-Net++ and HoverNet models over 3 runs across the various pre-trained weights for EoE, Crohns, PanNuke, and CoNSeP. We put the maximum performance for each test set and model across the pre-trained weights in bold and put a star if optimal performance is statistically significant. Notably, the models pre-trained on histopathology and the models pre-trained on ImageNet do not have differences that are statistically significant, except for HoVer-Net pre-trained on MoNuSAC for PanNuke, where $p = 0.052499$ from a Welch's t-test comparing it with the ImageNet pre-trained model performance. This indicates that pre-training on these histopathology datasets does not increase the segmentation performance relative to ImageNet weights. The randomly initialized weights are lower for all datasets except U-Net++ for EoE, suggesting that some kind of pre-training is useful. Furthermore, the number of epochs trained when using a model initialized with ImageNet weights is comparable to models pre-trained on histopathology, being significantly larger only for U-Net++ on PanNuke. Thus, there is no set of consistently optimal pre-trained weights, and the ImageNet weights provide the same or better performance than weights from a model pre-trained on multiple histopathology datasets. Also, the time for training for models with ImageNet pre-trained encoder is comparable to models pre-trained with histopathology.

Table 1: Model Performance

| Model | Pre-Trained Weights | Crohns Dice | Epochs | EoE Dice | Epochs |
|---|---|---|---|---|---|
| | | **Test Dataset** | | | |
| U-Net++ | Random | $0.55 \pm 0.033$ | 84 | **0.62** $\pm 0.009$ | 83 |
| U-Net++ | ImageNet | **0.572** $\pm 0.006$ | 31 | $0.615 \pm 0.02$ | 60 |
| U-Net++ | MoNuSAC | $0.554 \pm 0.009$ | 109 | $0.612 \pm 0.015$ | 103 |
| U-Net++ | CoNSeP | $0.565 \pm 0.019$ | 30 | $0.618 \pm 0.018$ | 63 |
| U-Net++ | PanNuke | $0.554 \pm 0.031$ | 11 | $0.599 \pm 0.001$ | 65 |
| U-Net++ | EoE | $0.557 \pm 0.026$ | 21 | - | - |
| U-Net++ | Crohns | - | - | $0.606 \pm 0.016$ | 89 |
| HoVer-Net | Random | $0.389 \pm 0.192$ | 74 | $0.572 \pm 0.007$ | 80 |
| HoVer-Net | ImageNet | **0.609** $\pm 0.012$ | 90 | $0.621 \pm 0.004$ | 93 |
| HoVer-Net | MoNuSAC | $0.6 \pm 0.011$ | 83 | **0.624** $\pm 0.002$ | 97 |
| | | **PanNuke** | | **CoNSeP** | |
| U-Net++ | Random | $0.552 \pm 0.014$ | 73 | $0.65 \pm 0.011$ | 129 |
| U-Net++ | ImageNet | $0.571 \pm 0.013$ | 127 | $0.669 \pm 0.008$ | 82 |
| U-Net++ | MoNuSAC | $0.54 \pm 0.012$ | 69 | $0.667 \pm 0.009$ | 109 |
| U-Net++ | CoNSeP | $0.57 \pm 0.02$ | 50 | - | - |
| U-Net++ | PanNuke | - | - | **0.678** $\pm 0.012$ | 62 |
| U-Net++ | EoE | **0.593** $\pm 0.033$ | 114 | $0.656 \pm 0.039$ | 83 |
| U-Net++ | Crohns | $0.577 \pm 0.017$ | 92 | $0.664 \pm 0.022$ | 81 |
| HoVer-Net | Random | $0.555 \pm 0.007$ | 78 | $0.403 \pm 0.048$ | 67 |
| HoVer-Net | ImageNet | $0.585 \pm 0.003$ | 94 | $0.67 \pm 0.007$ | 97 |
| HoVer-Net | MoNuSAC | **0.602**\* $\pm 0.008$ | 98 | **0.679** $\pm 0.016$ | 83 |
| HoVer-Net | CoNSeP | $0.589 \pm 0.003$ | 95 | - | - |
| HoVer-Net | PanNuke | - | - | $0.644 \pm 0.05$ | 78 |

## 4. Conclusion

We showed that training a model with ImageNet pre-trained weights did not have significantly different performance than pre-training on multiple histopathology datasets for 2 state-of-the-art medical segmentation models, the U-Net++ and HoVer-Net. This is likely due in part to the relatively small size of the datasets used in pre-training. Small datasets do not allow the model to learn a diversity of features, even when they come from the target domain. Furthermore, the number of training epochs to minimize the validation loss did not increase for the models pre-trained with ImageNet relative to those trained on histopathology. We conclude that, unless an abundant amount of histopathology data is available, pre-training on relatively small histopathology datasets is not likely to increase performance or decrease training time relative to an ImageNet baseline.

## Acknowledgments

This was supported by the National Center for Advancing Translational Science of the NIH Award UL1TR003015/ KL2TR003016 and NIH NIDDK K23 Award 5K23DK117061-03.

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
