# OpenReview forum: "Pre-training Segmentation Models for Histopathology"
_MIDL.io/2023/Short_Paper_Track — MIDL 2023 Short paper track Poster_

### Official Review · Reviewer_u9Rr · 2023-04-24
**Exploration of pre-training efficiency in histopathology, potential of discussions is good.**

**Rating:** 9
**Confidence:** 5

**Review:**

113 Pre-training segmentation models for histopathology

This abstract summarizes results on weither imagenet pretraining remains sufficient to segment histopathological images. Several methods (hovernet and unet++) and initialization schemes (datasets) are explored. Results indicate that imagenet pre-training is sufficient for most cases when compared to histopathology-based pre-training. Potential of discussions is good. Recommendation towards Acceptance.

---

### Official Review · Reviewer_8tNK · 2023-04-25
**useful comparison, solid study**

**Rating:** 7
**Confidence:** 3

**Review:**

This paper investigates whether pre-training on histopathology datasets can improve performance on histopathology (nuclei segmentation) tasks, in comparison to models that are pre-trained on natural images. Here, pre-training on a (smaller set of) histopathology data was not found to provide advantages over the use of ImageNet.

-	This study provides useful information about the utility of pre-training on medical images; but as the authors acknowledge, the lack of a benefit may be due to the smaller sample size of the histopathology data.

-	The study appears well-conducted and thorough

-	The novelty of the present work beyond the study of Ray et al. 2022 (mentioned in the Intro) may not be clear to readers, and it could be useful to expand upon that.